# Antimicrobial Activity and Chemical Composition of Essential Oils against Pathogenic Microorganisms of Freshwater Fish

**DOI:** 10.3390/plants10071265

**Published:** 2021-06-22

**Authors:** Alīna Klūga, Margarita Terentjeva, Nenad L. Vukovic, Miroslava Kačániová

**Affiliations:** 1Institute of Food and Environmental Hygiene, Faculty of Veterinary Medicine, Latvia University of Life Sciences and Technologies, K. Helmaņa iela 8, LV-3004 Jelgava, Latvia; vm09023@llu.lv (A.K.); margarita.terentjeva@llu.lv (M.T.); 2Department of Chemistry, Faculty of Science, University of Kragujevac, P.O. Box 12, 34 000 Kragujevac, Serbia; nvukovic@kg.ac.rs; 3Department of Fruit Sciences, Viticulture and Enology, Faculty of Horticulture and Landscape Engineering, Slovak University of Agriculture, Tr. A. Hlinku 2, 94976 Nitra, Slovakia; 4Department of Bioenergy, Food Technology and Microbiology, Institute of Food Technology and Nutrition, University of Rzeszow, Zelwerowicza St. 4, 35601 Rzeszow, Poland

**Keywords:** antimicrobial resistance, *Pseudomonas* spp., *Aeromonas* spp., *Yersinia* spp., disc diffusion method, Minimum Inhibitory Concentration (MIC)

## Abstract

Antimicrobials are widely applied in aquaculture for treatment of infectious diseases in fish. The increased antimicrobial resistance of fish pathogens to conventional antimicrobial treatment highlights the need for research on the antibacterial properties of natural products—in this case, essential oils (EOs). The aim of the present study was to detect the antimicrobial activity of the essential oils on pathogenic microorganisms found in freshwater fish. Freshwater fish isolates of *Aerococcus* spp., *Aeromonas* spp., *Enterococcus* spp., *Escherichia* spp., *Pseudomonas* spp., *Shewanella* spp., *Yersinia* spp., and *Vagococcus* spp. were tested for antimicrobial resistance and antimicrobial activity against 14 commercially available essential oils. Antimicrobial resistance was identified in *Pseudomonas* spp. isolates against cefepime and ciprofloxacin; while all *Aeromonas*, *Enterococcus*, and *Yersinia* isolates were fully susceptible. All tested EOs revealed antimicrobial activity against the tested freshwater fish isolates at different extents. *Cinnamomum camphora* exhibited strong antimicrobial activity against *Aeromonas* spp. (3.12 μL/mL), *Enterococcus* spp. (0.78–1.56 μL/mL), and *Pseudomonas* spp. with the MIC method. EOs of *Gaultheria procumbens* and *Litsea cubeba* showed strong antibacterial activity against *Yersinia* spp. and *Vacococcus* spp. (6.25 μL/mL). The study shows the antimicrobial activity of EOs against the most relevant freshwater fish pathogens and indicates the application opportunities in aquaculture.

## 1. Introduction

Essential oils are present in different parts of plants and consist of aromatic and volatile compounds. The primary role of EOs is the protection of plants against pathogens, which is attributed to the antimicrobial activity that EOs have shown [1]. The effects of EOs are created by their chemical composition, and an amount of a single compound from different plants, sharing different chemotypes affects the chemical composition of EOs [2]. Application of EOs may lead to alterations in the cell structure, e.g., the degradation of the cytoplasmic membranes and cell wall, and the synthesis of membrane lipids. EOs have been described as regulating the quorum sensing systems by the formation of biofilms and expression of virulence factors [3]. 

EOs are widely applied in cosmetics, perfumes, and food production because of their strong smells and flavors, and microbial growth inhibiting properties [4]. Different features of EOs have been revealed to have antimicrobial properties, which have been intensively investigated to find out their possible applications for replacements of existing antimicrobial treatments. The antibacterial, antiviral, antifungal, and insecticidal properties of EOs have been demonstrated, which allow us to consider EOs as an alternative of the current antimicrobials used for humans and animals [5,6,7]. 

Antimicrobial resistance is a global threat to animal and public health that affects the availability of antimicrobials for treatment of bacterial infections in animal and humans [8]. Fish pathogens may cause outbreaks with high mortality, leading to significant economic damage [9]. In aquaculture, the elevated stress, increased fish density, and deteriorated animal welfare may facilitate the rapid spread of fish diseases. Treatment with antimicrobials is needed to address this condition with application of antimicrobials to water and feed, which may result in ineffective therapeutic concentration and treatment of uninfected fish, which in turn may increase antimicrobial resistance [5,10]. 

Reduction in consumption of antimicrobials in human and veterinary medicine may minimize the problems related to the spread of antimicrobial resistance. Previous reports showed the antimicrobial activity of EOs against different microorganisms, including human and animal pathogens, from the perspective of applications in aquaculture [6,7,11,12]. EOs could be a promising agent in fish health, since their antimicrobial and immunomodulating properties have been described. Antimicrobial effects against such different fish pathogens and contaminants as *Aeromonas* spp., *Enterobacter* spp., *Pseudomonas* pp., *Vibrio* spp., etc. have been recorded with therapeutic application of EOs as recommended previously [11,12,13,14]. However, the diversity of fish microbiota and chemical composition of Eos, as well as the different antimicrobial effects of the EOs tested indicates the need for further research into these antimicrobial activities. Therefore, the aim of the present study was to investigate the antimicrobial activities of 14 commercially available EOs on the pathogens of freshwater fish.

## 2. Results

### 2.1. Chemical Composition of EOs

The chemical compositions of the tested essential oils are shown in Table 1. Elemol was one of the dominant compounds in EO of *Amyris balsamifera* L. and *Canarium luzonicum* Miq.(H) with 11.55% and 20.8% correspondingly. Additionally, τ-cadinol (33.4%) and 11-en-4-α-ol (11.3%) were found as main components in EO of *Amyris balsamifera* L. One of the major compounds in EOs of *Boswellia carterii* L. and *Cinnamomum camphora* Nees & Eberm was α-pinene—with levels of 37.0% and 12.2% respectively. The compound p-cimene (6.3%) was one of the major components found in EO of *Boswellia carterii* L. We found α-limonene to be the dominant constituent in EOs of *Boswellia carterii* L., *Canarium luzonicum* Miq.(H), *Cinnamomum camphora* Nees & Eberm, *Litsea cubeba* Pers., *Melaleuca leucadendron* L., and *Citrus limon* (L.) with 19.8%, 39.7%, 25.1%, 14.3%, 6.9%, and 58.9% respectively. In EO of *Canarium luzonicum* Miq.(H), α-phellandrene (12.6%) was one of the main compounds. The compound 1,8-cineole was detected as one of the dominant components in EOs of *Cinnamomum camphora* Nees & Eberm, *Melaleuca leucadendron* L., and *Melaleuca ericifolia* Smith, with 35.2%, 49.0%, and 16.9% correspondingly. Linalool (98.1%) was the main compound in EO of *Cinnamomum camphora* var. *linaloolifera*, with no other components present at a level of more than 1%. Linalool acetate (48.5%), linalool (22.8%), and α-terpineol (6.6%) were the main compounds in EO of *Citrus aurantium* L. Methyl salicylate was the main compound in EO of *Gaultheria procumbens* L. with 97.6%. Neral (22.8%) and geranial (22.8%) were found as the main components in EO of Litsea cubeba Pers. Aromadendrene (5.3%) was found as one of the major compounds in EO of Melaleuca ericifolia Smith. Patchouli alcohol (31.0%), α-bulnesene (21.3%), and α-guaiene (14.3%) were the main components in EO of *Pogostemon cabli* L. The compounds β-pinene (13.3%) and γ-terpinene (11.2%) were found in EO of *Citrus limon* (L.). We found that α-santalol (59.0%), α-bergamotene (9.68%), and β-santalol (9.02%) were the major compounds in EO of *Santalum album* L. Finally, β-vetivenene (7.42%), khusenol (5.24%) and β-guaiene (4.43%) were the main components in EO of *Vetiveria zizanoides* (L.) Roberty.

### 2.2. Antimicrobial Resistance in Fish Isolates 

Antimicrobial resistance in *P. fluorescens*, *P. frederiksbergensis*, and *P. gessardii* against cefepime and ciprofloxacine was found, while *P. lundensis* showed antimicrobial resistance only against cefepime and *P. proteolitica* only against ciprofloxacin. Antimicrobial resistance in *Aeromonas* spp., *Enterococcus* spp., and *Yersinia* spp. was not identified (Table 2). 

### 2.3. Antimicrobial Activity of Fish Isolates

All tested EO exhibited antimicrobial activity against microbial isolates (Table 3). *Cinnamomum camphora* var. *linalolifera* showed the strongest antimicrobial activity against *Aerococcus* spp. (20.33 ± 0.58 mm), *A. viridans* (19.67 ± 1.53 mm), *Aeromonas* spp. (15.33 ± 0.58 mm), *A. bestiarum* (16.67 ± 0.58 mm), *Enterococcus moravensis* (20.33 ± 1.53 mm), and *E. faecium* (17.67 ± 0.58 mm). 

*Cinnamomum camphora* Nees & Eberm expressed the strongest activity against A. salmonicida (17.33 ± 2.08 mm), *E. faecium* (19.33 ± 1.15 mm) and *E. aquimarinus* (19.00 ± 1.00 mm). 

*Amyris balsamifera* exhibited the strongest antimicrobial activity against *Pseudomonas proteolitica* (15.33 ± 0.58 mm), *Yersinia enterocolitica* (15.00 ± 1.00 mm) and *Yersinia* spp. (15.00 ± 1.00 mm). *Litsea cubeba* revealed the strongest activity against *Shewanella baltica* (14.33 ± 0.58 mm), *Y. ruckeri* (14.33 ± 0.58 mm), and *Vagococcus* spp. (14.33 ± 0.58 mm). *Santalum album* demonstrated the strongest antimicrobial activity against *P. flourescens* (10.67 ± 1.15 mm); but *Vitiveria zizanoides* against *Eschericia vulgaris* (10.67 ± 0.58 mm), *Pseudomonas fluorescens* (10.67 ± 1.15 mm), *P. frederiksbergensis* (10.67 ± 1.15 mm), *P. gessardii* (10.33 ± 0.58 mm), and *P. lundensis* (11.00 ± 1.00 mm).

*Cinnamomum camphora* exhibited the strongest antimicrobial activity against *Aerococcus* spp., *A. viridance* (0.156 µL/mL), *E. moravensis*, *E. faecium* (0.78 µL/mL), *A. bestiarum*, *A. salmonicida* (3.12 µL/mL), *E. faecium* (1.56 µL/mL), and *E. aquimarinus* (0.78 µL/mL), alongside *Cinnamomum camphora* Nees & Eberm. *Cinnamomum camphora* Nees & Eberm and *Amyris balsamifera* were the most active against *P. frederiksbergenis* (12.5 µL/mL). *Vetiveria zizanoides* showed antimicrobial activity against *P. gessardii* (12.5 µL/mL) and *P. lundensis* (6.25 µL/mL). *Gaultheria procumbens* and *Litsea cubeba* Pers. expressed the strongest antimicrobial activity against *Shewanella baltica*, *Y. ruckeri*, *Vagococcus* spp., and *Yersinia* spp.; they were also strongest alongside *Amyris balsamifera*, *Boswelia carterii*, *Malaleuca ericifolia*, *Citrus auarantum*, and *Malaleuca leucadendron* for *Y. enterocolitica* (6.25 µL/mL) (Table 4). 

## 3. Discussion

Among the chemical compositions of the EOs in the present study, several compounds, including α-pinene, β-pinene, α-limonene, α-terpineol, and 1,8-cineole have been reported previously. Studies on the antimicrobial activities of the isomers and enantiomers of pinene showed that α-pinene and β-pinene had antibacterial activity against *Cryptococcus neoformans*, *Candida albicans*, *Rhizopus oryzae*, and MRSA [15]. Limonene was found to have high antibacterial activity against Gram-positive and Gram-negative foodborne pathogens—*E. coli*, *S. enterica*, and *S. aureus* [16]. One of the major compounds in the leaf essential oils, α-terpineol, possessed the strongest antibacterial activities compared with the other components [17].

The antimicrobial effect of the EO of *Cinnamomum* spp. is attributed to the chemical composition of the plant. In the present study, 1,8-cineole and a-terpineol were found to be the main constituents of the EO, in agreement with Bottoni et al. [18]. Excellent antibacterial activities of 1,8-cineole was reposted in *S. aureus* and *E. coli*, associated with damage with cell compounds confirmed with electron microscopy [17].

In the present study, *Aeromonas* spp. was susceptible to all antimicrobials, while *Pseudomonas* spp. exhibited resistance to cefepime and ciprofloxacin. High rates of antimicrobial resistance in *Pseudomonas* spp. and low in Aeromonas spp. were in agreement with previous studies [19,20]. Susceptibility of *Enterococcus* spp. to antimicrobials was in line with Ellis–Iversen et al. [21], who found that the majority of *E. faecalis* and *E. faecium* were fully susceptible against antibiotics, while multiresistant *Enterococcus* was isolated from the Mediterranean aquaculture site and the fish-rearing ponds in Bangladesh without history of enterococcal infection [22,23]. *Yersinia* spp. found in pigs exhibited resistance to chloramphenicol, ciprofloxacin, and cephalosporins, and the ability of *Y. ruckeri* to develop antimicrobial resistance to quinolones was identified [24,25]. In general, the rates of antimicrobial resistance were low in the present study and may be attributed to low consumption of antimicrobials in aquaculture. 

All tested EOs exhibited antimicrobial activity against tested fish pathogens, with *Cinnamommum camphora* the most active against the majority of isolates. The strong antimicrobial activity of *Cinnamomum capphora* was identified previously for both Gram-positive and Gram-negative bacteria, with the MIC for *E. faecalis* being 1.6 mg/mL [26]. *Cinnamomum zeylanicum* was the most potent against *Pseudomonas* spp., both with the disc diffusion and MIC methods in fish isolates from Latvia [19]. The low activity of *C. camphora* against *A. salmonicida* subsp. *salmonicida* strains with MIC above 3200 μg/mL found in the study Hayaygheib et al. [27], was in contrast with our results. In our view, differences in the antimicrobial activity of EOs are attributed to the chemical compositions of EOs, with *Cinammomum capmhora* var. *linaloolifera* proving the most active against *Aeromonas* spp. and *Entecococcus* in the present study.

*Malaleuca alternifolia* expressed high activity against *A. hydrophila* and could be a natural alternative for prevention and control of the pathogen [13]. *M. alternifolia* exhibited antimicrobial activity against *A. hydrophila* isolates with the MIC method [28]. The EO of *M. alternifolia* is a well-known agent for local application, with antibacterial properties shown clinically [29]. The antimicrobial activity against *A. salmonicida* subsp. *salmonicida* was low in the Hayaybheib et al. study [27], which concluded that limonene and linalool showed a weak antimicrobial activity.

EO of *Vetiveria zizanioides* mostly expressed a weak antimicrobial activity. Orchard et al. [30], found that the EO of *Vetiveria zizanioides* was highly effective against *P. aeruginosa*. However, EOs from the roots of *V. zizanioides* and *V. nigritana* showed a low activity against Gram-negative bacilli but strong against Gram-positive cocci, in agreement with our results [31].

*Amyris balsamifera* showed antimicrobial activity against Gram-negative *Aeromonas* spp. and *Yersinia* spp. Antimicrobial activity of sandalwoods was reported previously, with high activity against *Klebsiella pneumonia* identified in the study of Jirovetz et al. [32]. High antimicrobial activity of sandalwood oil against Gram-positive *S. aureus* was observed, while different results were observed against *E. coli* and *Ps. aeruginosa* [32].

*Gaultheria procumbens* and *Litsea cubeba* showed antimicrobial activity against *Yersinia enterocolitica*. *G. procumbens* EO was more effective against Gram-negative bacteria than against Gram-positive bacteria; while high antimicrobial activity of *L. cubeba* against *Y. enterocolitica* was reported by Ebani et al. [33]—(15.7 ± 0.6 mm) in poultry isolates [34]. *Tripolum pannonicum* and *Origanum vulgare* exhibited antimicrobial activity against *Y. ruckeri* [35]. EO of *G. procumbens* and *L. cubeba* were active against *Sh. baltica*, which is recognized as an H_2_S producer in ice-stored fish from the Danish Baltic sea [36]. *Shewanella* may serve as an opportunistic pathogen of human and aquatic animals [37].

Despite the weak antimicrobial activity of *Citrus aurata* identified in the present study, the EOs of citrus were active against human and fish pathogenic strains, able to develop antibiofilm properties previously [38,39]. However, resistance or weak inhibition against *Escherichia* spp. and *Klebsiella* spp. of citrus EO in the studies of Moreira et al. [40] and Mancuso et al. [38] were in line with our results. 

## 4. Materials and Methods

### 4.1. Essential Oils

Altogether 14 essential oils (Hanus a.s., Nitra, Slovakia) were used in the present study: *Amyris Balsamifera* L., *Boswellia carterii* Birdw., *Canarium luzonicum* (Blume) A. Gray, *Cinnamomum camphora* (L.) J. Presl., *Cinnamomum camphora* var. *linaloolifera* Y. Fuita, *Citrus* x *aurantium* L., *Gaultheria procumbens* L., *Litsea cubeba* (Lour.) Pers., *Melaleuca eicifolia* Smith., *Melaleuca leucadendra* L., *Pogostemom cablin* (Blanco) Benth., *Citrus limon* (L.) Osbeck, *Santalum album* L., and *Vitiveria zizanoides* (L.) Roberty.

### 4.2. Chemical Characterization of Essential Oil Samples by Gas Chromatography/Mass Spectrometry (GC/MS) and Gas Chromatography (GC-FID)

GC/MS analyses of selected essential oil samples were performed using an Agilent 6890N gas chromatograph (Agilent Technologies, Santa Clara, CA, USA) coupled to quadrupole mass spectrometer 5975B (Agilent Technologies, Santa Clara, CA, USA). A HP-5MS capillary column (30 m × 0.25 mm × 0.25 μm) was used. The temperature program was: 60 °C to 150 °C (increasing rate 3 °C/min) and 150 °C to 280 °C (increasing rate 5 °C/min). The total run time was 60 min. Helium 5.0 was used as the carrier gas with a flow rate of 1 mL/min. The injection volume was 1 μL (essential oil samples were diluted in pentane), while the split/splitless injector temperature was set at 280 °C. With a split ratio at 40.8:1, investigated samples were injected in the split mode. Electron-impact mass spectrometric data (EI-MS; 70 eV) were acquired in scan mode over the m/z range 35–550. MS ion source and MS quadrupole temperatures were 230 °C and 150 °C, respectively. Acquisition of data started after a solvent delay time of 3 min. GC-FID analyses were performed on Agilent 6890N gas chromatograph coupled to FID detector. Column (HP-5MS) and chromatographic conditions were same as for GC-MS. FID detector temperature was set at 300 °C. The individual volatile constituents of injected essential oil samples were identified based of their retention indices [41] and comparison with reference spectra (Wiley and NIST databases). The retention indices were experimentally determined using the standard method [42], which included retention times of n-alkanes (C6-C34), injected under the same chromatographic conditions. The percentages of the identified compounds (amounts higher than 0.1%) were derived from their GC peak areas.

### 4.3. Microbial Isolates

Fish isolates were originated from wild and aquacultured roach (*Rutilus rutilus*) and bream (*Abramis brama*) from skin, gills, and gut samples. Wild fish were caught in freshwaters (lake and river) in Latvia, while aquacultured fish were bought from producers after their placement on the market. Microbial isolates were confirmed with Maldi-TOF MS Biotyper (Bruker Daltonics, Bremen, Germany). Isolates of fish health and public health significance were selected for the present study with the following genera were represented: *Aerococcus* spp., *Aeromonas* spp., *Escherichia* spp., *Enterococcus* spp., *Pseudomonas* spp., *Shewanella* spp., *Yersinia* spp., *Vagococcus* spp.

### 4.4. Detection of Antimicrobial Resistance

Antimicrobial resistance was tested with the disc diffusion method with the following antimicrobials selected: cefepime (FEP, 30 µg), ciprofloxacin (CIP, 5 µg), levofloxacin (LEV, 5 µg), chloramphenicol (C, 30 µg), imipenem (IMP, 10 µg), teicoplanin (TEI, 30 µg), tigecycline (TGC, 15 µg), linezolid (LZD, 10 µg), tobramycin (TOB, 10 µg) (Oxoid, Basingstone, UK). Interpretation of results was done according to EUCAST [43].

### 4.5. Detection of Antimicrobial Activity with Disc Diffusion Method

A 0.1 mL of suspension of tested culture (10^5^ cfu/mL) was used for inoculation of Mueller Hinton Agar (MHA, Oxoid, Basingstoke, UK). Blank discs were impregnated with 15 µL of EO and placed onto inoculated agar. The agars were incubated at 4 °C for 2 h then at 30 °C and 37 °C for 24 h. The zone of inhibition was measured and the zone < 10 mm was accepted as not inhibitory. All analyses were done in triplicate.

### 4.6. Detection of Minimum Inhibitory Concentration

Tests were performed according to the Clinical and Laboratory Standards Institute (CLSI) in Mueller Hinton Broth (MHB, Oxoid, Basingstoke, UK). After overnight incubation at 30 °C and 37 °C, a bacterial suspension of 10^6^ cfu/mL was used for inoculation. A 96-well micro-titer plate was used with 50 μL added to each well, excluding the 10th well where 100 μL was added for sterility control. The EO solution in dimethyl sulphoxide (DMSO, Penta, Prague, Czech Republic) was used for inoculation, with 5% of DMSO added to the 10th well. Suspensions were done by transferring of 50 μL to each next well. MIC values were determined by measuring of turbidity. The MIC values was expected to be the lowest concentration of the EO inhibiting bacterial growth. The test was performed in triplicate, and cefoxitin (30 μL) was used as positive control.

### 4.7. Data Analysis

Mean, standard deviations for the antimicrobial activities of EOs were calculated. One-way analysis of variance (ANOVA) was applied to detect differences for antimicrobial effects of different EOs on microorganisms.

## 5. Conclusions

Identified antimicrobial resistance in *Pseudomonas* spp. isolates shows the emergence of the spread of antimicrobial resistance in aquaculture. Despite the antimicrobial resistance in *Aeromonas* isolates being lower than reported previously, there is evidence of the presence of resistant isolates in wild isolates of freshwater fish. All EOs tested in the present study exhibited antimicrobial activity against wild isolates of freshwater fish, with EO of *Cinnamomum camphora* Nees & Eberm, and *Cinnamomum caphora* var. *Linalolifera* proving the most active against *Aerococcus* spp., *Aeromonas* spp., and *Enterococcus* spp. EOs of *Gaultheria procumbens* and *Litsea cubeba Pers.* exhibited the strongest antimicrobial activity against *Shewanella* spp., *Yersinia* spp., and *Vagococcus* spp. Efficiency of EO against foodborne pathogens may be a promising strategy to prevent and/or treat infectious diseases in fish in a sustainable way by reducing the consumption of antimicrobials in aquaculture. Potential applications of EOs and antimicrobial activity of specific compounds in EOs against fish pathogens need to be further studied.

## Figures and Tables

**Table 1 plants-10-01265-t001:** Chemical composition of essential oils (%) *.

Essential Oil	Components ^a^	Percentage of Components ^b^
*Amyris balsamifera* L.	elemol	11.5
	τ-cadinol	33.4
	selin-11-en-4-α-ol	11.3
*Boswellia carterii* L.	α-pinene	37.0
	*p*-cimene	6.3
	α-limonene	19.8
*Canarium luzonicum* Miq.(H)	α-phellandrene	12.6
	α-limonene	39.7
	elemol	20.8
*Cinnamomum camphora* Nees & Eberm	α-pinene	12.2
	α-limonene	25.1
	1,8-cineole	35.2
*Cinnamomum camphora* var. *linalolifera*	linalool	98.1
*Citrus aurantium* L.	linalool	22.8
	α-terpineol	6.6
	linalool acetate	48.5
*Gaultheria procumbens* L.	methyl salicylate	97.6
*Litsea cubeba* Pers.	α-limonene	14.3
	neral	29.5
	geranial	39.4
*Melaleuca leucadendron* L.	α-limonene	6.9
	1,8-cineole	49.0
	α-terpineol	7.8
*Melaleuca ericifolia* Smith.	1,8-cineole	16.9
	linalool	47.5
	aromadendrene	5.3
*Pogostemon cabli* L.	α-guaiene	14.3
	α-bulnesene	21.3
	patchouli alcohol	31.0
*Citrus limon* (L.)	β-pinene	13.3
	α-limonene	58.9
	γ-terpinene	11.2
*Santalum album* L.	α-bergamotene	9.68
	α-santalol	59.0
	β-santalol	9.02
*Vetiveria zizanoides* (L.)	β-vetivenene	7.42
	β-guaiene	4.43
	khusenol	5.24

Note: ***** listed are the main components, **^a^** Identified compounds, **^b^** compounds identified in amounts. Full composition of chemical compounds is shown in Appendix A.

**Table 2 plants-10-01265-t002:** Antimicrobial resistance of fish isolates.

Pathogen	Antimicrobial, Inhibition Zone (in mm)								
	FEP	CIP	LEV	C	IMP	TEI	TGC	LZD	TOB
*Aerococcus* spp.	ND (25)	ND (31)	ND (25)	ND (24)	NT	NT	NT	NT	NT
*Aerococcus viridans*	ND (26)	ND (28)	ND (25)	ND (30)	NT	NT	NT	NT	NT
*Aeromonas* spp.	S (30)	S (30)	S (30)	NR (29)	NT	NT	NT	NT	NT
*Aeromonas bestiarum*	S (29)	S (28)	S (31)	ND (32)	NT	NT	NT	NT	NT
*Aeromonas salmonicida*	S (28)	S (30)	S (35)	ND (30)	NT	NT	NT	NT	NT
*Escherichia vulgaris*	S (30)	S (27)	S (25)	S (28)	NT	NT	NT	NT	NT
*Enterococcus faecium*	NT	NT	NT	NT	S (25)	S (24)	S (25)	S (30)	NT
*Emterococcus moravensis*	NT	NT	NT	NT	S (26)	S (23)	S (28)	S (25)	NT
*Enterococcus faecium*	NT	NT	NT	NT	S (21)	S (20)	S (25)	S (22)	NT
*Enterococcus aquimarinus*	NT	NT	NT	NT	S (22)	S (25)	S (27)	S (22)	NT
*Pseudomonas fluorescens*	R (48)	R (45)	S (60)	NT	NT	NT	NT	NT	S (25)
*Pseudomonas frederiksbergensis*	R (52)	R (51)	S (51)	NT	NT	NT	NT	NT	S (20)
*Pseudomonas gessardii*	R (47)	R (51)	S (56)	NT	NT	NT	NT	NT	S (20)
*Pseudomonas lundensis*	R (45)	S (53)	S (45)	NT	NT	NT	NT	NT	S (23)
*Pseudomonas proteolitica*	S (53)	R (25)	S (55)	NT	NT	NT	NT	NT	S (28)
*Shewanella baltica*	ND (30)	ND (27)	ND (25)	ND (30)	NT	NT	NT	NT	NT
*Yersinia enterocolitica*	S (28)	S (30)	S (28)	S (29)	NT	NT	NT	NT	NT
*Yersinia ruckeri*	S (30)	S (30)	S (28)	S (31)	NT	NT	NT	NT	NT
*Yersinia* spp.	S (28)	S (27)	S (28)	S (32)	NT	NT	NT	NT	NT
*Vagococcus* spp.	ND (30)	ND (25)	ND (28)	ND (30)	NT	NT	NT	NT	NT

Abbreviations: FEP–cefepime, CIP–ciprofloxacin, LEV–levofloxacin (LEV), C–chloramphenicol, IMP–imipenem, TEI–teicoplanin, TGC-tigecycline, LZD–linezolid, TOB–tobramycin, ND–not determined, NT–not tested, S–sensitive, R-resistant.

**Table 3 plants-10-01265-t003:** Antimicrobial activity of essential oils with the disc diffusion method.

Pathogen	Essential Oil, Zone of Inhibition in mm ± SD													
	1	2	3	4	5	6	7	8	9	10	11	12	13	14
*Aerococcus* spp.	13.67 ± 0.58	14.33 ± 0.58 ^c^	14.66 ± 0.58 ^a,d^	16.00 ± 1.00	20.33 ± 0.58 ^a,f^	9.00 ± 1.00	9.67 ± 1.15	13.00 ± 1.00	15.33 ± 0.58 ^j^	13.67 ± 0.58	12.67 ± 0.58	10.67 ± 0.58	11.33 ± 0.58	12.33 ± 0.58
*Aerococcus viridans*	14.67 ± 0.58	12.67 ± 1.15	10.67 ± 1.15 ^a^	15.67 ± 0.58	19.67 ± 1.53 ^a,f^	10.67 ± 0.58	11.33 ± 1.15	11.33 ± 0.58	12.67 ± 1.15	11.0 ± 1.0	8.33 ± 0.58	8.33 ± 0.58	8.33 ± 0.58	12.33 ± 0.58
*Aeromonas* spp.	11.67 ± 0.58	13 ± 1.73	11.67 ± 0.58	18 ± 1.0 ^e^	15.33 ± 0.58	12.67 ± 1.15 ^g^	17.67 ± 0.58 ^h^	12.67 ± 1.15	10.33 ± 0.58	12.67 ± 0.58	15.00 ± 1.00	11.33 ± 0.58	14.33 ± 1.09	12.00 ± 1.00
*Aeromonas bestiarum*	15.33 ± 0.58 ^b^	13.00 ± 1.00	8.67 ± 1.15 ^a^	15.33 ± 0.58	16.67 ± 0.58 ^a^	11.33 ± 0.58	15.00 ± 0.00 ^h^	13.00 ± 1.73	12.33 ± 0.58	11.33 ± 1.15	10.00 ± 1.00	11.33 ± 1.15	11.33 ± 0.58	10.33 ± 1.53
*Aeromonas salmonicida*	11.67 ± 0.58	12.33 ± 0.58	11.67 ± 0.58	17.33 ± 2.08 ^e^	15.00 ± 0.00	12.00 ± 1.00	17.33 ± 1.15 ^h^	12.00 ± 2.00	11.00 ± 0.00	12.33 ± 0.58	14.33 ± 0.58	12.33 ± 0.58	13.67 ± 2.3	12.00 ± 0.00
*Escherichia vulgaris*	9.33 ± 0.58 ^b,k^	7.00 ± 1.00 ^c,k^	7.00 ± 1.73 ^d,k^	7.67 ± 0.58 ^e,k^	7.00 ± 1.73 ^f,k^	7.33 ± 1.15 ^k^	7.67 ± 0.58 ^h,k^	8.33 ± 0.58 ^k^	10.00 ± 1.00	9.67 ± 0.58	10.33 ± 0.58	8.67 ± 1.15	10.33 ± 0.58	10.67 ± 0.58
*Enterococcus faecium*	14.67 ± 0.58	13.67 ± 0.58 ^c^	12.33 ± 0.58 ^a^	19.33 ± 1.15 ^e^	18.67 ± 1.15 ^a^	12.00 ± 0.00	13.67 ± 1.15	14.33 ± 0.58 ^i^	11.67 ± 0.58	11.67 ± 0.58	12.33 ± 0.00	12.00 ± 0.00	10.33 ± 1.53	9.67 ± 0.58
*Enterococcus moravensis*	15.00 ± 1.00 ^b^	13.33 ± 0.58 ^c^	11.00 ± 1.00 ^a^	16.00 ± 1.00	20.33 ± 1.53 ^a,f^	12.00 ± 1.00	14.67 ± 0.58	13.67 ± 1.53	12.67 ± 0.58	12.00 ± 2.00	11.00 ± 1.00	11.67 ± 0.58	11.67 ± 0.58	11.00 ± 1.00
*Enterococcus faecium*	15.33 ± 0.58 ^b^	14.67 ± 0.58 ^c^	12.33 ± 0.58 ^a^	15.33 ± 0.58	17.67 ± 0.58 ^a^	12.00 ± 0.00	13.67 ± 1.15	14.33 ± 0.58 ^i^	11.67 ± 0.58	11.67 ± 0.58	12.33 ± 0.58	12.00 ± 0.00	10.33 ± 1.53	9.67 ± 0.58
*Enterococcus aquimarinus*	15.00 ± 1.00 ^b^	13.33 ± 0.58 ^c^	11.00 ± 1.00 ^a^	19.00 ± 1.00 ^e^	21.00 ± 1.00 ^a,f^	12.00 ± 1.00	14.67 ± 0.58	13.67 ± 1.53	12.67 ± 0.58	12.00 ± 2.00	11.00 ± 1.00	11.67 ± 0.58	11.67 ± 0.58	11.00 ± 1.00
*Pseudomonas fluorescens*	10.00 ± 1.00 ^k^	7.33 ± 1.53 ^k^	7.00 ± 1.73 ^d,k^	7.67 ± 1.15 ^e,k^	6.00 ± 1.00 ^f,k^	8.00 ± 1.00 ^k^	8.00 ± 1.00 ^k^	8.33 ± 1.53 ^k^	9.33 ± 1.15 ^j^	10.00 ± 1.00	9.33 ± 0.58	9.67 ± 1.53	10.67 ± 1.15	10.67 ± 0.58
*Pseudomonas frederiksbergensis*	9.67 ± 0.58 ^b,k^	7.00 ± 1.00 ^c,k^	6.67 ± 1.53 ^d,k^	8.00 ± 1.73 ^k^	6.00 ± 1.00 ^f,k^	7.33 ± 1.15 ^k^	7.67 ± 0.58 ^h,k^	8.33 ± 0.58 ^k^	9.67 ± 0.58	10.00 ± 1.00	10.00 ± 0.00	8.67 ± 0.58	10.33 ± 0.58	10.67 ± 1.15
*Pseudomonas gessardii*	9.33 ± 0.58 ^b,k^	7.00 ± 1.00 ^c,k^	6.33 ± 1.53 ^d,k^	7.67 ± 1.15 ^e,k^	6.00 ± 1.00 ^f,k^	8.00 ± 1.00 ^k^	8.00 ± 1.00 ^k^	8.33 ± 0.58 ^k^	9.33 ± 0.58 ^j^	9.67 ± 0.58	10.00 ± 1.00	9.00 ± 1.00	9.33 ± 0.58	10.33 ± 0.58
*Pseudomonas ludensis*	9.33 ± 1.15 ^b,k^	6.67 ± 0.58 ^c,k^	6.67 ± 1.53 ^d,k^	7.67 ± 1.15 ^e,k^	6.00 ± 1.00 ^f,k^	7.00 ± 1.00 ^g,k^	7.67 ± 1.15 ^h,k^	7.33 ± 1.15 ^i,k^	9.00 ± 1.00 ^j^	9.00 ± 1.00	10.00 ± 1.00	8.67 ± 1.15	10.33 ± 1.53	11.00 ± 1.00
*Pseudomonas proteolitica*	15.33 ± 0.58 ^b^	13.00 ± 1.00	8.67 ± 1.15	9.00 ± 1.00	11.00 ± 1.00	11.33 ± 0.58	15.00 ± 0.00 ^h^	13.00 ± 1.73	12.33 ± 0.58	11.33 ± 1.15	10.00 ± 1.00	11.33 ± 0.58	11.33 ± 0.58	10.33 ± 1.53
*Shewanella baltica*	11.00 ± 1.00	13.00 ± 1.00	11.33 ± 1.15	10.67 ± 1.15	11.67 ± 0.58	12.00 ± 0.00	13.67 ± 1.15	14.33 ± 0.58 ^i^	11.67 ± 0.58	11.67 ± 0.58	12.33 ± 0.58	12.00 ± 0.00	10.33 ± 1.53	9.67 ± 0.58
*Yersinia enterocolitica*	15.00 ± 1.00 ^b^	13.33 ± 0.58 ^c^	11.00 ± 1.00	10.33 ± 0.58	11.33 ± 1.15	12.00 ± 1.00	14.67 ± 0.58	13.67 ± 1.53	12.67 ± 0.58	12.00 ± 2.00	11.00 ± 1.00	11.67 ± 0.58	11.67 ± 0.58	11.00 ± 1.00
*Yersinia ruckeri*	11.00 ± 1.00	13.00 ± 1.00	11.33 ± 1.15	10.67 ± 1.15	11.67 ± 0.58	12.00 ± 0.00	13.67 ± 1.15	14.33 ± 0.58 ^i^	11.67 ± 0.58	11.67 ± 0.58	12.33 ± 0.58	12.00 ± 0.00	10.33 ± 1.53	9.67 ± 0.58
*Yersinia* spp.	15.00 ± 1.00 ^b^	13.33 ± 0.58 ^c^	11.00 ± 1.00	10.33 ± 0.58	11.33 ± 1.15	12.00 ± 1.00	14.67 ± 0.58	13.67 ± 1.53	12.67 ± 0.58	12.00 ± 2.00	11.00 ± 1.00	11.67 ± 0.58	11.67 ± 0.58	11.00 ± 1.00
*Vagococcus* spp.	11.00 ± 1.00	13.00 ± 1.00	11.33 ± 1.15	10.67 ± 1.15	11.67 ± 0.58	12.00 ± 0.00	13.67 ± 1.15	14.33 ± 0.58 ^i^	11.67 ± 0.58	11.67 ± 0.58	12.33 ± 0.58	12.00 ± 0.00	10.33 ± 1.53	9.67 ± 0.58

Essential oils–1. *A. balsamifera*, 2. *B. carterii*, 3. *Canarium luzonicum* Miq.(H), 4. *Cinnamomum camphora* Nees & Eberm, 5. *Cinnamomum camphora* var. *linalolifera*, 6. *Citrus auarantium*, 7. *Gaultheria procumbens*, 8. *Litsea cubeba* Pers., 9. *Melaleuca leucadendron*, 10. *Malaleuca ericifolia* Smith., 11. *Pogostemon cabli*, 12. *Citrus limon*, 13. *Santalum album*, 14. *Vetiveria zizanoides*. ^a^ There were significant differences between the antimicrobial activity of *Cinnamomum camphora* var. *Linalolifera* and *Canarium luzonicum* on *Aerococcus* spp., *A. viridans*, *A. bestiarum*, *E. moravensis*, *E. faecium*, *E. faecium*, and *E. aquimarinus*. *Canarium luzonicum* shows significantly stronger antibacterial effects (*p* < 0.05). ^b^
*A. balsamifera*’s antibacterial effect on *A. bestiarum*, *E. moravensis*, *E. faecium*, *E. faecium* and *E. aquimarinus*, *P. proteolitica*, *Y. enterocolitica*, and *Yersinia* spp. was significantly higher than on *E. vulgaris*, *P. frederiksbergensis*, *P. gessardii*, and *P. ludensis* (*p* < 0.05). ^c^
*B. carterii* shows a significantly higher antibacterial effect on *Aerococcus* spp., *Enterococcus moravensis*, *Enterococcus faecium*, *Enterococcus aquimarinus*, *Yersinia enterocolitica*, and *Yersinia* spp. than on *Escherichia vulgaris*, *Pseudomonas frederiksbergensis*, *Pseudomonas gessardii*, and *Pseudomonas ludensis* (*p* < 0.05). ^d^
*Canarium luzonicum* shows a significantly stronger antibacterial effect on *Aerococcus* spp. than on *E. vulgaris*, *P. fluorescens*, *P. frederiksbergensis*, *P. gessardii* and *P. ludensis* (*p* < 0.05). ^e^
*Cinnamomum camphora* shows a significantly higher antibacterial effect on *Aeromonas* spp., *Aeromonas salmonicida*, *E. faecium* and *E. aquimarinus* than on *E. vulgaris*, *P. fluorescens*, *P. gessardii* and *P. ludensis* (*p* < 0.05). ^f^
*Cinnamomum caphora* var. *Linalolifera* shows a significantly stronger antibacterial effect on *Aerococcus* spp., *A. viridans*, *E. moravensis* and *E. aquimarinus* than on *E. vulgaris*, *P. fluorescens*, *P. frederiksbergensis*, *P. gessardii* and *P. ludensis* (*p* < 0.05). ^g^ There is a significant difference between the antimicrobial activity of *Citrus auarantium* on *Aeromonas* spp. and *P. ludensis* (*p* < 0.05). ^h^
*Gaultheria procumbens* shows a significantly higher antibacterial effect on *Aeromonas* spp., *A. bestarium*, *A. salmonicidum* and *P. proteolitica* than on *E. vulgaris*, *P. frederiksbergensis* and *P. ludensis* (*p* < 0.05). ^i^
*Litsea cubeba* Pers. shows a significantly lower antibacterial effect on *P. ludensis* than on *E. faecium*, *Shewanella baltica*, *Yersinia ruckeri* and *Vagococcus* spp. (*p* < 0.05). ^j^
*Melaleuca leucadendron* shows a significantly higher antibacterial effect on *Aerococcus* spp. and *Vagococcus* spp. than on *P. fluorescens*, *P. gessardii* and *P. ludensis* (*p* < 0.05). ^k^ There is no significant difference between the antimicrobial activity of EOs on *E. vulgaris*, *P. fluorescens*, *P. frederiksbergensis*, *P. gessardii* and *P. ludensis* (*p* > 0.05).

**Table 4 plants-10-01265-t004:** Antimicrobial activity of essential oils tested with the Minimum Inhibitory Concentration (MIC) method.

Pathogen	Essential Oil, MIC μL/mL													
	*A. balsamifera*	*B. carterii*	*C. luzonicum Miq.(H)*	*Cinnamomum camphora* Nees & Eberm	*Cinnamomum camphora* var. *Linalolifera*	*Citrus auarantium*	*Gaultheria procumbens*	*Litsea cubeba pers.*	*Melaleuca leucadendron*	*Malaleuca ericifolia smith*	*Pogostemon cabli*	*Citrus limon*	*Santalum album*	*Vetiveria zizanoides*
*Aerococcus* spp.	12.5	6.25	3.12	3.12	1.56	25.0	12.5	12.5	6.25	6.25	12,5	12.5	12.5	6.25
*Aerococcus viridans*	3.12	6.25	12.5	1.56	1.56	12.5	6.25	12.5	12.5	25.0	50.0	50.0	50.0	25.0
*Aeromonas* spp.	12.5	12.5	12.5	3.12	3.12	25.0	3.12	12.5	12.5	6.25	3.12	6.25	1.56	12.5
*Aeromonas bestiarum*	3.12	6.25	50.0	3.12	3.12	12.5	6.25	6.25	6.25	6.25	12.5	6.25	12.5	25.0
*Aeromonas salmonicida*	12.5	12.5	12.5	3.12	3.12	12.5	3.12	12.5	12.5	12.5	6.25	12.5	3.12	12.5
*Escherichia vulgaris*	25.0	25.0	25.0	25.0	50.0	50.0	25.0	25.0	12.5	12.5	12.5	25.0	12.5	12.5
*Enterococcus faecium*	3.12	3.12	6.25	1.56	1.56	6.25	3.12	3.12	6.25	6.25	6.25	6.25	6.25	6.25
*Enterococcus moravensis*	3.12	3.12	6.25	1.56	0.78	6.25	3.12	3.12	6.25	6.25	12.5	6.25	6.25	12.5
*Enterococcus faecium*	3.12	3.12	6.25	1.56	0.78	6.25	3.12	3.12	6.25	6.25	6.25	6.25	6.25	12.5
*Enterococcus aquimarinus*	3.12	3.12	6.25	0.78	0.78	3.12	3.12	3.12	3.12	1.56	6.25	6.25	6.25	6.25
*Pseudomonas fluorescens*	12.5	25.0	25.0	25.0	25.0	25.0	25.0	25.0	12.5	12.5	12.5	25.0	25.0	25.0
*Pseudomonas frederiksbergensis*	12.5	25.0	25.0	12.5	25.0	25.0	25.0	25.0	50.0	50.0	50.0	25.0	25.0	25.0
*Pseudomonas gessardii*	25.0	25.0	50.0	25.0	50.0	25.0	25.0	25.0	12.5	12.5	12.5	25.0	12.5	12.5
*Pseudomonas ludensis*	25.0	25.0	50.0	25.0	25.0	50.0	25.0	12.5	12.5	12.5	12.5	25.0	12.5	6.25
*Pseudomonas proteolitica*	6.25	12.5	25.0	25.0	25.0	25.0	6.25	12.5	12.5	12.5	25.0	12.5	25.0	25.0
*Shewanella baltica*	12.5	12.5	12.5	25.0	12.5	12.5	6.25	6.25	12.5	12.5	12.5	12.5	125	25.0
*Yersinia enterocolitica*	6.25	6.25	12.5	12.5	12.5	6.25	6.25	6.25	6.25	6.25	25.0	12.5	25.0	25.0
*Yersinia ruckeri*	12.5	12.5	12.5	25.0	12.5	12.5	6.25	6.25	12.5	12.5	12.5	12.5	12.5	25.0
*Yersinia* spp.	6.25	6.25	25.0	25.0	25.0	12.5	6.25	6.25	12.5	6.25	12.5	12.5	25.0	25.0
*Vagococcus* spp.	12.5	12.5	12.5	25.0	12.5	12.5	6.25	6.25	12.5	12.5	12.5	12.5	12.5	25.0

## Data Availability

Data is contained within the article.

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
