# Peer review of "Antimicrobial Activity and Chemical Composition of Essential Oils against Pathogenic Microorganisms of Freshwater Fish"

_plants, 2021, doi:10.3390/plants10071265_

Round 1
Reviewer 1 Report
The article “Antimicrobial activity and chemical composition of essential 2 oils against pathogenic microorganisms of freshwater fish” was submitted for review.
Essential oils (EOs) are present in various parts of plants and have antimicrobial activity. The antiviral, antifungal and insecticidal properties of essential oils have been demonstrated, which make it possible to consider EO as an alternative to antimicrobial drugs used for humans and animals. Fish pathogens can cause outbreaks with high mortality resulting in significant damage. Reducing the consumption of antimicrobial drugs in human and veterinary medicine can minimize the problems associated with the spread of antimicrobial resistance. Thus, the aim of this study was to study the antimicrobial biological activity of commercially available EOs on pathogenic microorganisms of freshwater fish.
The introduction contains information about the relevance of the problem under study, the need to study it. The Results section contains information on Chemical composition of EOs, Antimicrobial resistance in fish isolates, Antimicrobial activity of fish isolates. The Discussion section contains comparison of data with known results, contains literature references. The Results section has information on essential oils, chemical characteristics of essential oil samples, Detection of Antimicrobial Resistance, Detection of Antimicrobial Activity with Disc Diffusion Method, Detection of Minimum Inhibitory Concentration, Data analysis. The article is illustrated with 3 tables.
Conclusion. The article is a full-fledged study, well stated, it is recommended for publication without changes.
Author Response
Reviewer #1
The article “Antimicrobial activity and chemical composition of essential 2 oils against pathogenic microorganisms of freshwater fish” was submitted for review.
Essential oils (EOs) are present in various parts of plants and have antimicrobial activity. The antiviral, antifungal and insecticidal properties of essential oils have been demonstrated, which make it possible to consider EO as an alternative to antimicrobial drugs used for humans and animals. Fish pathogens can cause outbreaks with high mortality resulting in significant damage. Reducing the consumption of antimicrobial drugs in human and veterinary medicine can minimize the problems associated with the spread of antimicrobial resistance. Thus, the aim of this study was to study the antimicrobial biological activity of commercially available EOs on pathogenic microorganisms of freshwater fish.
The introduction contains information about the relevance of the problem under study, the need to study it. The Results section contains information on Chemical composition of EOs, Antimicrobial resistance in fish isolates, Antimicrobial activity of fish isolates. The Discussion section contains comparison of data with known results, contains literature references. The Results section has information on essential oils, chemical characteristics of essential oil samples, Detection of Antimicrobial Resistance, Detection of Antimicrobial Activity with Disc Diffusion Method, Detection of Minimum Inhibitory Concentration, Data analysis. The article is illustrated with 3 tables.
Conclusion. The article is a full-fledged study, well stated, it is recommended for publication without changes.
We thank the Reviewer for their positive comment and careful review.

Reviewer 2 Report
The ms on essential oils against pathogenic microorganisms of freshwater fish is interesting, well written (except of some structural organisation, clarity of tables), and contains a large amount of data.
I wonder why the submitted ms with such issues was not submitted to the journals dealing with the problems of aquaculture or veterinary fish farming?
The Title is well constructed.
The Abstract should be shortened, to present the most important results, giving the strongest effect of essential oils on selected bacteria. I would recommend avoiding in this paragraph (and for other ones too) so many species Latin names.
Keywords: why these two bacteria species are given?
The Introduction is written correctly and contains the most important information about the ms topic.
The results should be rewritten and tables need reorganization for increasing the clarity of the text and the presentation of the results. Table 1 is very long and detailed, presenting many substances, I recommend converting the table into an annex and attach it to the end of the ms. By the way, did the Authors try to correlate the quantitative data of substances extracted from the essential oils (% content) with antimicrobial activity (multifactor statistical analyses are useful)?
Table 2" Antimicrobial resistance of fish isolates" in paragraph 2.2 needs correction for clarity, the pathogen names should be abbreviated, and the graphical structure simplified.
The next Table on “Antimicrobial activity of essential oils with disc diffusion method” should have probably the number 3. This Table is too big and too detailed to see the results (20 pathogens x 14 oil types including mean value with SD). I think it would be reconstructed to be more “friendly” for readers converting the big Table into the graphical plot (e.g. bars with SD).
Table no 3 (4? ms line 171) with results of MCI method should be also converted into a graphical plot (see my comments to the previous table) to increase visibility.
The scientific impact of the paper would be greater if the data, many variables, and their relationships were analyzed by more advanced statistical methods.
The Methods described properly.
Conclusions are short; the Authors should stress the importance of further studies on essential oils, and a need to test specific components of oils and their impact on fish pathogens.
Author Response
Reviewer #2
The ms on essential oils against pathogenic microorganisms of freshwater fish is interesting, well written (except of some structural organisation, clarity of tables), and contains a large amount of data.
I wonder why the submitted ms with such issues was not submitted to the journals dealing with the problems of aquaculture or veterinary fish farming?
Microbial activity of EOs is multidiscipline research broadly covered by many science areas including chemistry, food sciences, veterinary medicine etc. Authors think that the present topic could be interesting for broad auditorium including professionals and researchers in plants.
The Title is well constructed.
The Abstract should be shortened, to present the most important results, giving the strongest effect of essential oils on selected bacteria. I would recommend avoiding in this paragraph (and for other ones too) so many species Latin names.
The content of abstract is shortened to 200 words according to the Journal requirements. Since common names for plants and microorganisms could be misleading the scientific names in abstract were used characterization of specific microorganisms and plants.
Keywords: why these two bacteria species are given?
These bacteria genera were highlighted in keywords because of their importance for fish health. Included bacteria also pose public health concerns and alter the quality and safety of fish meat. These keywords may be helpful for authors from different areas to track the present manuscript.
The Introduction is written correctly and contains the most important information about the ms topic.
The results should be rewritten, and tables need reorganization for increasing the clarity of the text and the presentation of the results. Table 1 is very long and detailed, presenting many substances, I recommend converting the table into an annex and attach it to the end of the ms. By the way, did the Authors try to correlate the quantitative data of substances extracted from the essential oils (% content) with antimicrobial activity (multifactor statistical analyses are useful)?
Table 1 is shortened, and explanatory text was rewritten. Detailed information on the composition of essential oil was added as the Supplemental Material 1. Results were rewritten for Table 1 for more clear explanation of results.
Table 2" Antimicrobial resistance of fish isolates" in paragraph 2.2 needs correction for clarity, the pathogen names should be abbreviated, and the graphical structure simplified.
Tables 2 was simplified to show the main pattern of antimicrobial resistance of microorganisms in the present study.
The next Table on “Antimicrobial activity of essential oils with disc diffusion method” should have probably the number 3. This Table is too big and too detailed to see the results (20 pathogens x 14 oil types including mean value with SD). I think it would be reconstructed to be more “friendly” for readers converting the big Table into the graphical plot (e.g., bars with SD).
Table no 3 (4? ms line 171) with results of MCI method should be also converted into a graphical plot (see my comments to the previous table) to increase visibility.
Numeration of tables was corrected. The construction of the graphical plot is a good option for better visibility of the results, however, due to diversity of microorganisms and EOs we could not find the appropriate solution for graphical representation of the results. Additionally, the inhibition value is important for characterization of antimicrobial activity of EO, therefore we think that the tables will be a better option for presentation of the results.
The scientific impact of the paper would be greater if the data, many variables, and their relationships were analysed by more advanced statistical methods.
We agree that more sophisticated methods for data analysis are available. However, the values of antimicrobial activity were analysed to detect the differences between more than two groups in unrelated samples because of characteristics of the obtained data.
The Methods described properly.
Conclusions are short; the Authors should stress the importance of further studies on essential oils, and a need to test specific components of oils and their impact on fish pathogens.
Sentence on the importance of the further studies was added to “Conclusions” section.

Reviewer 3 Report
- Materials and Methods: 14 essential oils in used in this study were all extracted in plants? The extraction method and voucher number and authentication of their plant source should be presented.
- Materials and Methods: To test antimicrobial activity, EO treatment to agar was carried out. When EO treatment to agar, method of EO preparation as test materials should be written.
- Antimicrobial activity of each main component of EO should be added.
- Add discussion about antimicrobial activity of each main compound of EO.
Author Response
Reviewer #3
Materials and Methods: 14 essential oils in used in this study were all extracted in plants? The extraction method and voucher number and authentication of their plant source should be presented.
The samples of EO produced in Slovakia (Hanus s.r.o.) were used in the experiment. The extraction procedures of plants were not done therefore the procedure was not included in the manuscript.
Materials and Methods: To test antimicrobial activity, EO treatment to agar was carried out. When EO treatment to agar, method of EO preparation as test materials should be written.
Pure EO without additional supplements was used for impregnation of paper disc for detection of antimicrobial activity as described in 4.4. subsection.
Antimicrobial activity of each main component of EO should be added.
Antimicrobial activity of each main component was not tested in the present study because EO usually consists of mixture of different compounds. The present study shows antimicrobial properties of essential oils available from producers.
Add discussion about antimicrobial activity of each main compound of EO.
Discussion about the main compounds of EO was added.

Round 2
Reviewer 3 Report
Revised form is acceptable.